# Exploring registered nurses' perspectives as mentors for newly qualified nurses: a qualitative interview study

Stina Kallerhult Hermansson [ID] ,[1] Anne Kasén,[2] Yvonne Hilli,[2]
Fredrik Norström [ID] ,[3] Jonas Rennemo Vaag,[2] Karin Bölenius[1]

**To cite:** Kallerhult
Hermansson S, Kasén A, Hilli Y,
*et al.* Exploring registered
nurses' perspectives as
mentors for newly qualified
nurses: a qualitative
interview study. *BMJ Open*
2024;**14**:e082940. doi:10.1136/
bmjopen-2023-082940

► Prepublication history
and additional supplemental
material for this paper are
available online. To view these
files, please visit the journal
online (https://doi.org/10.1136/
bmjopen-2023-082940).

¹Department of Nursing, Umeå
University, Umeå, Sweden
²Faculty of Nursing and Health
Sciences, Nord University, Bodö,
Norway
³Department of Epidemiology
and Global Health, Umeå
University, Umeå, Sweden

**Correspondence to**
Mrs Stina Kallerhult
Hermansson;
stina.hermansson@umu.se

## ABSTRACT

**Objective** Existing research has focused mostly on
mentees' experiences of mentoring rather than mentors'
experiences. Therefore, this study describes registered
nurses' experiences of being a mentor for newly qualified
nurses.

**Design** A qualitative interview study based on
semistructured individual interviews. Interviews were
analysed using qualitative content analysis.

**Participants and setting** A purposive sample of
experienced registered nurses (n=21) from healthcare
units in northern Sweden and northern Norway. Inclusion
criteria were to have been a mentor to at least one
newly qualified nurse, hold permanent employment
of 75%–100% as a registered nurse and to be able to
communicate in Swedish or Norwegian.

**Results** Our study's findings suggest that being a mentor
plays a crucial role in establishing safety in complex
work environments. The main theme consists of three
themes: feeling motivated in being a mentor; continuously
developing the learning environment; and navigating
obstacles and cultivating support.

**Conclusion** Being a mentor is a complicated role for
registered nurses. The mentoring role is beneficial—ie,
positive and rewarding—if facilitated sufficiently in the
context of a structured organisation. This study brings a
more profound understanding of and provides new insights
into registered nurses' perspectives and needs regarding
being a mentor and the study's findings make an important
contribution to the field of nursing regarding the facilitation
of mentoring.

## INTRODUCTION

Mentoring is beneficial and has proven to be
vital for the longevity of a registered nurse's
working life. It contributes to the retention
of registered nurses and supports registered
nurses' professional development.[1–3] Never-
theless, implementing mentoring in health-
care organisations is challenging. Literature
has shown that a variety of factors contribute
to these challenges. For example, there is
a lack of standardisation and shared views
on mentoring in previous research and in

## STRENGTHS AND LIMITATIONS OF THIS STUDY

⇒ The study employed data triangulation to enhance
credibility, and included participants of diverse ages,
genders and organisational contexts across munic-
ipalities and counties, contributing to a comprehen-
sive understanding.
⇒ Investigator triangulation was a notable strength,
involving multiple researchers in the interview pro-
cess, data analysis and interpretation, ensuring ro-
bustness and reliability in the study's findings.
⇒ The use of an interview guide enabled uniformity in
the interview process, enhancing the dependability
of the study.
⇒ Reflexivity was addressed in the methods section,
in line with the goal of establishing confirmability,
ensuring awareness of potential biases and contrib-
uting to the study's methodological rigour.
⇒ Transferability was promoted by providing detailed
descriptions of the study's context and participants;
however, consideration for international perspec-
tives should acknowledge healthcare organisational
diversities across different countries as a potential
limitation.

clinical practice. High workload, organisa-
tional factors and high staff turnover are
other contributing factors.[4–6] The benefits
and implications of mentoring are known.
However, research is sparse, and little is known
about mentors' perspectives. It is reasonable
to believe that this knowledge gap contrib-
utes to why most mentoring implementations
have been short-term and unsuccessful.[5]

The literal meaning of 'mentoring' is
defined in the Cambridge Dictionary as 'the
activity of supporting and advising someone
with less experience to help them develop in
their work'.[7] The actual meaning of mento-
ring is more complex than the literal defini-
tion. Mentoring in nursing can mean many
different things, and the concept is used in
various ways.[8] Preceptorship and mentorship

are often intertwined and are often used synonymously.[5 9] A mentor can be defined in various ways,[10] in the present study, we define a 'mentor' as an experienced registered nurse that has been a contact person and support for the newly qualified nurse during their transition and introduction to the workplace. We define a 'mentee' as a newly qualified nurse receiving mentoring.

Mentoring can be a dynamic and positive relationship consisting of honesty, close connection and trust between the mentor and the mentee.[11] Previous studies suggest that from the mentees' point of view, an ideal mentor is a nurse with seniority who is open to communication and feedback; someone who can provide a safe learning environment; and an individual with knowledge, skills and a positive attitude.[12 13] Successful mentors are approachable and support professional socialisation by promoting a sense of belonging. Mentors counsel, actively listen and encourage development. An effective mentoring relationship in nursing empowers the mentee.[12 13] Thus, mentoring is needed on several aspects of a new registered nurse's career. Regardless, mentoring programmes tends to focus on orientation of the everyday routines and on clinical training.[10 14] A recent review by Hoover et al[10] showed that the frequency of mentoring is often up to the mentor and mentee to decide, and the duration can be from 3 months to more than a year. The mentor-mentee ratio can be anything from 1:1 to 1:5.[10] However, statistics on mentor to mentee ratios, frequency of contact, duration of mentoring, etc. is scarce. An integrative review by Hampton et al[15] concluded that there is a need for greater standardisation of mentoring concepts in nursing. Also, research on the mentors' point of view and what they need to be able to be this ideal kind of mentor is limited.

The fundamental nature of mentoring consists of multiple support roles and a longer duration than, for example, a traditional introduction, ie, mentoring can be ongoing with no precise end date, since support is needed continuously.[16] In addition to being a supportive relationship, mentoring can be defined as an organisational intervention to promote the growth and socialisation of newly registered nurses in an organisation.[17] Although mentoring can be defined in various ways, most studies agree that mentoring is relational and developmental, has career and psychosocial functions, and involves phases and transitions.[13] The phases can range, for example, from discovery and testing to reciprocal interactions; in the end, the mentee perceives themselves as equal to their mentors regarding their professional capability.[11]

Dall 'Alba[18] explained this transition to a professional role in terms of becoming. Personal growth and professional development occur in stages that rely on constructive relationships and support from other professionals, for example, mentors. Becoming a professional is more than obtaining the knowledge and skills from an education. 'Becoming' is a learning process that transforms the whole person while their professional identity is being discovered. Learning professional ways of being involves the integration of knowing, acting and being the professional in question.[18]

In summary, the benefits and implications of mentoring and the qualities of an ideal mentoring relationship are known—from the mentee's perspective. A scoping review by Aldosari et al[19] showed a lack of qualitative research regarding the experiences and perceptions of mentors in nursing practice. The present study will help fill the knowledge gap of mentoring from the mentors' point of view. The aim of the present study is to describe registered nurses' experiences of being a mentor for newly qualified nurses.

## METHODS
### Design and context
A qualitative descriptive study design with individual interviews was chosen to obtain detailed descriptions of the participating mentors' experiences, in line with Brinkmann and Kvale.[20] The composition and reporting of this study were guided by The Standards for Reporting Qualitative Research.[21] This study is part of the overarching research programme 'Becoming a Professional Nurse (BePROF)', a comprehensive Nordic project aiming to develop an evidence-based mentorship programme to increase support for registered nurses in northern Sweden and northern Norway.

At the time of the interviews, the study participants worked as registered nurses in healthcare units in hospitals and municipalities in northern Sweden and northern Norway. The healthcare systems in the Nordic countries are organised similarly, although each organisational level has some degree of variation in the content and responsibilities. Hospital care is predominantly confined to state and county levels, and municipalities are responsible for home care services and elderly and disability care, eg, nursing homes.[22]

### Participants and sampling
Participants were selected through purposive sampling[23] to ensure inclusion of individuals with extensive knowledge of the subject matter, that is, the experiences of being a mentor. Seeking a diverse range of experiences, we welcomed mentors with varying backgrounds, numbers of mentees and different mentoring routines within their respective units. We observed variations in mentoring practices across units, both within and between countries. Thus, it was imperative to include registered nurses from diverse contexts, various hospital settings and even rural areas to establish a comprehensive and representative sample for an in-depth study of mentoring.

First, we sent an email invitation to the operational managers in all units (hospital units and municipal care units) in one northern county in each country. Operational managers from 13 Swedish and eight Norwegian units approved the invitation. Then, managers investigated interest and provided contact information to registered nurses willing to participate. The inclusion criteria

were to have been a mentor to at least one newly qualified nurse, hold permanent employment of 75%–100% as a registered nurse and to be able to communicate in Swedish or Norwegian. Lastly, we invited 25 registered nurses. The goal was to conduct at least 17 interviews, in line with research on saturation in qualitative research.[24] In this stage, four registered nurses in Sweden declined the invitation, and all the invited registered nurses in Norway accepted the invitation. In total, 21 registered nurses participated in the study.

## Data collection

The research team performed one-on-one semistructured interviews with the participants in January–May 2021 via the digital platforms Teams or Zoom. The participants chose where they were located during the digital interview; it could be at home or at work. The participants were not reimbursed for their time; however, we had an agreement with leaders in the participating units that the participants could partake in the interviews during working hours. An interview guide was developed by members of the research team who had experience of being a mentor. The interview guide was inspired by current evidence about mentoring. Reference groups in participating municipalities and counties reviewed the interview guide prior to the interviews, and a pilot interview was performed. The pilot interview confirmed that the interview questions generated answers about experiences of mentoring showing that the interview guide was valid. After consensus was reached by the research team, interview questions were established to be open-ended with reflective elements and focused on mentoring as being and becoming a mentor. Three main questions were asked: 'Can you tell me what it means to be a mentor?', 'Can you tell me about what you specifically focus on as a mentor?' and 'How would you like to be a mentor and develop the mentorship?'. Each question had follow-up questions, for example, 'What made you choose to become a mentor?'. General follow-up questions throughout the interview could be 'Can you give an example of…', 'Can you elaborate on…', etc. (See Interview guide (online supplemental file). The median interview time was 59 min. Data saturation was deemed sufficient after conducting 21 interviews of varying lengths, with the longest interview lasting 90 min. The interviews were perceived as spontaneous and rich in content, and the interviewer followed up on the answers to elicit more detailed narratives. Interviews were audio-recorded and transcribed verbatim. Transcriptions were anonymised and stored at a secure storage space; recordings were discarded.

## Data analysis

The qualitative content analysis, following the methodological approach described by Graneheim et al,[25 26] guided the analyses. Authors analysed both manifest and latent content. The process began with repeated readings of the text, followed by extraction of meaning units and their condensation to capture core meanings. These units were then coded by the first author (SKH), who also sorted and grouped codes with similar content. Subthemes were created by all authors, discussing the content until consensus was reached. The subthemes were thereafter abstracted into themes, illustrated in table 1. The analysis process was iterative, moving between text, codes, abstraction and writing, with frequent comparisons to the original text to avoid over-interpretation. Saturation was determined based on the emergence of little or no new relevant codes in the data, coupled with the repetition of issues, as described by Hennink and Kaiser.[24] A consensus on themes and results was reached through discussions among all authors.

## Ethical considerations

The Swedish Ethical Review Authority (Dnr 2020–06187) and the Norwegian Centre for Research Data (No. 148896) have approved the study. We followed the recommendations on ethical principles in human research recommendations described in the Declaration of Helsinki.[27] For example, participants were informed orally and in writing about the study, that participation was voluntary, and that they were free to decline to participate at any point during the study prior to publication. Data were handled with confidentiality, that is, no unauthorised person have access to any data material.

There are always risks involved in interviewing, due to the closeness in interview situations. In other words, there is a risk of the researcher being too close or too distant.[28] Although the interview questions in this study can be regarded as non-controversial, the risk described above was considered during the interviews. In addition, there was also a risk that taking part in the interviews

| **Condensed meaning units** | **Codes** | **Subthemes** | **Themes** |
|---|---|---|---|
| *I would like to be an example; I would like the person sitting next to me to also feel that I am a role model in attitude and care*. (I.5. SWE) | Being an example and role model | Striving to be a role model | Feeling motivated in being a mentor |
| *There are always scientific changes, and we have to be receptive to that and I think it is very important that the mentor also has the ability to be flexible*. (I.9.NOR) | Flexibility—be open to new knowledge | Facing new challenges | Continuously developing the learning environment |
| *I find it frustrating sometimes. That you don't get that time to sit down and go through things in peace*. (I.3.SWE) | Lack of time leads to frustration | Struggling with work organisation | Navigating obstacles and cultivating support |

**Table 1** Examples of the abstraction process

could increase participants' workload and/or occupy their personal time. This risk was reduced by the previously mentioned agreement with managers that participants were able to complete the interviews during their normal working hours. The risks and the benefits of the study have been taken into consideration, and the benefits have been estimated to outweigh the risks.

## Trustworthiness

Following Lincoln and Guba's evaluative criteria on establishing trustworthiness,[29] credibility was established through data triangulation, as participants were selected to encompass a variety of ages and genders, as well as differences in organisations/healthcare units across participating municipalities and counties. This approach included participants with diverse experiences, thereby enhancing the credibility of this study. Investigator triangulation (ie, peer-debriefing) reinforces the study's credibility. Multiple researchers conducted the interviews, analysed data and interpreted the results. Uniformity in interviews was ensured by employing the same interview guide, thereby enhancing the study's dependability. The study's context and participants are described in detail, in an effort to establish transferability. The study's transferability to an international perspective should, however, be carried out in consideration of the diversities of healthcare organisations in different countries. For ensuring confirmability, reflexivity is addressed by acknowledging that researchers involved in this study had no professional obligations to the participating units or pre-existing relationships with any registered nurses participating in the study. SKH, AK, YH and KB have backgrounds as registered nurses and experiences in mentoring and therefore had a pre-understanding about the issues addressed in this study. They have different specialties and experiences from different fields of nursing. The research team also consist of senior researchers in public health and work environments (FN and JRV).

## Patient and public involvement

Patients or public were not involved in the design or planning of the study. Reference groups in participating units had the opportunity to give their opinions on the interview guide.

## RESULTS

In total, 21 registered nurses participated in the study, from Sweden (n=13) and Norway (n=8). The participants were of various ages, genders and had a variety of work experiences from different geographic areas and healthcare units in each county. The median age was 43 years (range 31–59). Participants were from hospital care units (n=15) that included surgical, medical, neurological, oncology, neonatal and intensive care units. We also interviewed participants from primary healthcare and municipal care (n=6).

The results suggest that being a mentor plays a crucial role in establishing safety in complex work environments. The themes and subthemes are illustrated in table 2. The main theme reflects participants' experiences of creating safety for their mentees, while contributing to their own professional development, and of developing a learning environment in a complex and stressful work environment. Being a mentor was described as an unclear role that needed to be clarified and defined; however, they were still motivated to be a mentor.

Some aspects promoted mentoring, such as experiencing professional development and being a role model, that is, the first theme, 'feeling motivated in being a mentor'. The second theme— 'continuously developing the learning environment'—covered creating safety while facing challenges and adapting in mentoring. The third theme—'navigating obstacles and cultivating support'— summed up barriers such as lack of time and support from leaders, as well as enabling factors such as supportive colleagues.

The themes and subthemes are described in more detail in the following sections and verified with quotes from the participants.

## Feeling motivated in being a mentor

Participants expressed how they see it as their mission to support the mentee in becoming more secure in their professional role as a registered nurse. Participants were 'striving to be a role model', which they described as one of the most crucial parts of being a mentor. Being a role model was described as multifaceted and involved a combination of personal characteristics and professional competence. Participants also reflected on the reasons

**Table 2** Overview of results

| Subthemes | Themes | Main theme |
|---|---|---|
| Striving to be a role model | Feeling motivated in being a mentor | Being a mentor plays a crucial role in establishing safety in complex work environments |
| Fostering professional development | | |
| Facing new challenges | Continuously developing the learning environment | |
| Adapting mentoring based on the mentee's needs | | |
| Struggling with work organisation | Navigating obstacles and cultivating support | |
| Enabling support and collegiality | | |

why they are mentors. 'Fostering professional development' meant that their motivation, interest and goals promoted being a mentor.

*Striving to be a role model* was expressed in various ways by the participants; for example, they talked about how they want to show moral support and strived to be trustworthy. They also described that they strived to promote self-confidence in the mentee. The participants wanted to show competence and show that they were available as mentors. As mentors, they also aimed to show a sense of safety to support the mentee in challenging situations. They saw it as exemplary to be transparent and clear about the goals of the mentoring relationship.

The participants described various qualities they strived to have as role models. The experiences they possessed as registered nurses, including communication skills and vast knowledge, were part of what made them role models. Having experience was described as valuable in one's capacity to be a mentor. According to the participants, a mentor should be competent to enable sharing of knowledge and create a trusting relationship with the mentee. Having a permissive attitude was described as necessary. Other qualities of a good mentor included sensitivity, patience and empathy. They desired to be open and neutral as a mentor. According to the participants, a mentor should strive to be self-aware, self-reflective and convey calmness. Being a role model could consist of different aspects; for example, being there for the mentee, being a source of professional inspiration and being supportive. One of the participants described 'being supportive' as follows:

> For me, being a mentor means I'm there to support my new colleagues, to provide a bit more neutrality so that my fellow colleagues can feel safe in talking in general terms about their experiences in their nursing roles, about health and safety, about dealing with colleagues and other professional categories – being somebody that that person can turn to. (Interview 5, SWE)

The participants expressed that they wanted to inspire the mentee to be confident and secure in standing up for their decisions. By continuous mentoring, they saw that the mentee could find the professional competence and motivation to remain in the profession. In addition to showing the mentee how to be confident, participants also wanted to show the mentee that it is permissible to not know everything. It was described as essential not to steer but to guide, a mentor is someone who listens but does not have all the answers, as expressed in the following quote:

> Being a mentor means showing you don't have to know everything, but that you can still feel secure in the knowledge that you don't know everything. […] Providing moral support, secure in your professional role to the extent that you can convey a sense of security in your professional role. (Interview 1, SWE)

*Fostering professional development* meant that the participants were personally interested in supporting a new colleague. Being a mentor was part of the profession; they described it as 'coming with the territory'. Participants expressed that they appreciated the positive response they got from the mentees. When the mentee was interested in learning, participants were encouraged. The participants saw it as rewarding that mentoring promoted their own professional development. Growing in the profession was described as promoting job satisfaction. Mentoring created security and joy at work for both the mentor and the mentee, according to the participants. The participants expressed feelings of excitement in being a mentor. They felt they had the capacity and talent to be suitable to be a mentor and had a lot of experience and knowledge to offer. Mentors were sometimes selected by management, which could be flattering and motivating but could also jeopardise their interest and motivation as a mentor. The goal of being a mentor was to facilitate competent colleagues and ensure the professional development of the department, and mentoring could lead to the development of a sustainable work environment. Participants described it as 'quality assurance'. This participant shared how professional development in mentoring could benefit patient care:

> But sharing the knowledge born of experience is precisely what's so important if others are to succeed in their work, and also to achieve the best possible patient outcomes. So, I think exchanging experience is incredibly important. (Interview 15, NOR)

Regardless of how they became a mentor, the participants described factors that guided them in their mentoring. Being a mentor contributed to their professional development. Seeing mentees thrive in the profession was described as 'rewarding' and increased participant's own competence. Participants described that their competence was made visible during mentorship. A motivational factor in being a mentor was that some participants did not have a mentor when they themselves were new nurses, and they described that they missed having a mentor. In contrast, others described the drive to give the same support they had received as new nurses. Being interested in mentoring was essential to be a successful mentor, as described by this participant:

> But then I think most people could do a good job as mentors if they have an interest in it and the desire to do it. But I think you have to have an interest in it – that's the top priority. To be able to do a good job, you have to want to do it. (Interview 6, SWE)

### Continuously developing the learning environment

The participants described fast-changing healthcare, which places high demands on a learning environment that supports staff in continuous development. The participants described how new nurses, in their opinion, needed different support today than they did a few years

ago. 'Facing new challenges' meant that they, as mentors, needed to keep up with new and updated knowledge. The participants reflected that there was no 'best way' to be a mentor, they had to 'adapt their mentoring based on the mentee's needs' and knowledge base. A mentor was thought to be essential, regardless of other introductory efforts and routines. The mentees' individual needs guided how long the mentorship lasted and the content of the mentoring.

*Facing new challenges* meant that as mentors, they needed to have 'humility for the new', as described by the participants. Mentoring and support were needed not only in clinical situations but also in interactions with other professional categories. The participants tried to foster a permissive atmosphere, where mentees felt safe asking questions. The participants also saw a need for mentoring for all new nurses, but it could be challenging if the new nurses did not think they needed mentoring. It was also described as challenging not to 'give too much' but to encourage the mentee to develop their own abilities.

Participants saw talking about feelings as essential, and that the most challenging aspect was dealing with existential issues if the mentee, for example, had cared for a dying patient. There were challenges keeping up with the development of the current requirements placed on new nurses. The participants saw a need for a longer mentoring period than simply an introduction. They had to be receptive to new knowledge, as described by this participant:

> You feel you keep your wits about you when you're a mentor, that you constantly have to go on learning new things. And not least, you have to be sure that what you're teaching is correct and up to date on a professional level. (Interview 8, NOR)

*Adapting mentoring based on mentee's needs* meant that mentoring could vary based on the mentee's previous experiences. With a confident mentee, the mentor could be less 'strict'. However, with an insecure mentee, the mentor had to be more hands-on, according to the participants, and guide the mentee more regularly. Mentoring could be adapted to quick and irregular check-ins, as described by this participant:

> Perhaps dealing with mentoring 'on the hoof', so to speak, there and then, but then maybe also being able to sit down and give it time. You adapt to the situation. (Interview 17, SWE)

Several participants described a matching process for the mentee–mentor relationship, where they saw how personal chemistry affected the mentoring. The participants emphasised they needed to know the mentee to plan the mentorship, not only the mentee's previous work experience, but also family life, other obligations outside working hours, etc.

A mentoring conversation stimulates reflection and consideration, according to the participants. The participants tried to focus on the positive aspects of the mentee's

development. The ambition was to let the mentee control what was addressed in the discussions, based on the mentees' individual needs. The participants encouraged the mentee to prepare for the mentoring conversation. To be a mentor was also described by the participants as to guide in conflicts, for example if there was a disagreement with a colleague. To continuously follow-up on challenges in the job.

The mentoring conversations were based on confidentiality to enable openness. The participants expressed that the aim was to meet one another unconditionally. They thought it was better with freer frames and flexibility around the conversation. It could be anywhere, based on the conditions in the workplace, if it were in a trustworthy environment. For example, mentoring could be adapted into a short and spontaneous reflection after a work shift. Mentoring can develop the mentees' professional skills without resorting to criticism, according to the participants.

> But it depends entirely on who I'm mentoring, (the mentee) is the one who should decide what we're going to talk about. (Interview 2, SWE)

### Navigating obstacles and cultivating support

The participants expressed experiences of navigating obstacles in the work environment that could hinder being a mentor fully. They described 'struggles with work organisations', and they experienced pressure from leaders and colleagues. The participants also reflected on factors that cultivated mentorship in a supportive environment. They reported that mentoring was enabled if leaders were attentive, and the organisation was well-functioning. 'Enabling support and collegiality' was also expressed as a necessary part of a supportive environment. Participants also claimed that having structure and predictability in the working day helped them in planning mentoring activities, and that structure and predictability were essential elements of cultivating a supportive environment.

*Struggling with work organisation* meant that participants expressed obstacles related to and frustration with how leaders and management handled mentorship. Responsiveness and sensitivity of managers were lacking, according to the participants. Participants had experienced that they were handed responsibility but not power to influence. Participants requested clarification regarding what mentorship was supposed to be, to be able to navigate the mentor role. For example, this participant expressed thoughts of ending their mentorship due to unclear definitions of the role:

> Given that (the mentoring role) has been a bit woolly, I was almost starting to think I should give up mentoring because I have plenty of other things going on at work. […] Maybe things are getting a bit much, and if I had to turn anything down, then the mentoring would be the first thing to go. (Interview 7, SWE)

They said it was not beneficial for the work environment to 'be forced' to be a mentor. Participants also perceived mentoring to be a non-priority in the eyes of management. Mentoring was often de-prioritised for other tasks in the workplace.

It was challenging to be a mentor due to insufficient training in, for example, conversational techniques and motivational conversations. They expressed that 'mentoring in mentoring' and contact with other mentors were lacking. The will was there—but education and development in mentoring needed to be improved to be able to cultivate a supportive environment. The participants described experiences of staff shortages and high staff turnover that led to obstacles in mentoring. For example, participants had experiences of simultaneously mentoring several new nurses, which could lead to difficulties in navigating the time management and priorities.

They expressed a need for better planning and structure within the organisation. It was described as necessary with sensitivity between the workgroup and the mentee. However, there needed to be a more precise division of responsibilities for follow-up in the mentoring. They experienced pressure from colleagues and felt responsible for the mentee and their skills. It was described as challenging to mentor someone whom colleagues saw as an insecure mentee.

Difficulties in planning, such as scheduling factors, could be obstacles for mentoring; there needed to be more time set aside for mentoring. The lack of time led to frustration, and not having set aside time made learning difficult, according to the participants. They needed help preparing. For example, they prepared for mentoring outside working hours. They streamlined mentoring due to lack of time, which led to insufficient development in the mentee. The participants experienced difficulties in navigating mentoring activities during periods of high workload. However, they saw a greater need for mentoring during these periods. This frustration is expressed by one of the participants in this quote:

> The management must take the time to follow-up. No progress will be made unless they do. But they often feel there's no time for it, so it's not a priority. So, it's there, but it grinds to a halt. (Interview 14, NOR)

*Enabling support and collegiality* meant that a structured environment was essential for being a mentor. Receiving support from team leaders reduced the obstacles and improved the possibilities to enable mentorship. Good management and leadership led to personalised mentoring, according to the participants. When leaders made it possible for them to navigate in both informal and formal structures for mentorship, this was an example of an attentive leader. This is expressed in the following quote:

> I also have to make it clear that we have a good manager who pays attention to everyone and understands everyone. She's also really good at keeping things on

track. I've often felt that as our manager, she's the glue that bonds the group together. (Interview 12, NOR)

The participants stated that flexibility in scheduling and planning was a prerequisite for being a successful mentor. According to the participants, having time allocated for both mentoring and patient-related work was a prerequisite for cultivating support. Having time for reflection (less workload) also promoted being a mentor. The participants described that having a mentee could reduce but also increase the workload. They described alternating mentorship as preferable, having some space between mentorships and replacing each other during absences.

The participants expressed that it was important to be loyal to one another, even though they were all different and could do things differently. It was described as essential to receive support from other mentors (for example, to talk about obstacles and 'difficult cases'). Sharing the responsibility and supporting other mentors and colleagues were considered cultivating. Participants expressed a will to build a value base for mentoring, create a mentorship structure and navigate a plan together. They gave examples of mentor groups where they planned new and reflected on previous mentorships. They expressed that they needed more regular meetings to exchange experiences.

They also shared how everyone in the workgroup took part in the mentoring, but only some had the title 'mentor'. Discussions were requested between their colleagues in the workgroup on frameworks and responsibilities regarding mentorship. Participants expressed the importance of taking care of one another. According to the participants, everyone in the workgroup had to be prepared and welcoming to the mentee, and the social work environment was essential for the mentees' learning. The workplace had to be inclusive and responsive, according to the participants. Participants also expressed that supportive leaders and a supportive organisation were essential for them continuing to be a mentor. Attitudes and organisational culture were described as vital components in successful mentoring:

> But just like here, people need a management team that does the same thing. I can't be the only one, I rely on us having the same ideas about (mentoring). It's really important. (Interview 13, NOR)

## DISCUSSION

The principal findings are that the complexity of the mentor role was evident in the registered nurses' narratives, and they strived to establish safety for mentees in complex work environments. Results show that the mentor role lacked clarity and entailed positive as well as negative experiences. For example, participants talked about being motivated, continuously developing the

learning environment, and navigating obstacles and cultivating support.

## Feeling motivated being in the role of a mentor

The results of the study show that being a mentor includes emphasis on creating trusting relationships with their mentees and being able to share knowledge with one's mentees. These results are in accordance with previous studies that concluded that mentoring success depends on the quality of the mentoring relationships, ie, the personal connection between the mentor and the mentee.[30–32] Also, according to Dall 'Alba,[18] people cannot become professionals by themselves; they can fully develop through contact and integration with others. Participants in our study strived to be a role model and offer support to their new colleagues, while at the same time develop in their own new professional identity of being a mentor.

In addition to supporting their mentees' professional development, being a mentor was also described as beneficial for the participants' own personal and professional development, as confirmed in previous research.[31] The participants in our study expressed that being a role model, being trustworthy, and having competence and experience as a nurse were prerequisites for mentoring. This confirms previous descriptions of an ideal mentor from the mentees' point of view that emphasised similar characteristics.[12 13]

## Continuously developing the learning environment

The present study adds new perspectives to previous studies regarding mentorship. For example, a systematic review by Kakyo et al[31] concluded that the content of mentoring is vital for it to work. In contrast, our results showed that it was important to have freer frames and adapt mentoring based on mentees' needs. Our results showed that content was not the most important aspect of mentorship; the participants in our study emphasised more about *how* and *why* mentoring was performed in their narratives. To develop a learning environment, a need to build a value base for mentoring and create a mentorship structure together with other mentors and colleagues was shown.

The experiences of developing the learning environment and being a role model can be compared with how Dall 'Alba[18] describes 'learning to become a professional' as an interplay between 'what I am and what I can become'. This means inheriting a world of other professionals, as well as inventing one's own world as a professional. Our study shows similar findings. Being a mentor meant having a sense of security in their own professional role and instilling self-confidence in their new colleagues, so that they, in turn, can build their own professional identities. This is confirmative to Yip et al[33] who showed in their study that support from senior colleagues gave new nurses hope, and facilitated their self-transformative process in their professional development as registered nurses.

Other than getting to know their mentee and adapting to their personal learning needs, participants in the present study also expressed challenges in keeping up with the rapid development of new knowledge. A meta-synthesis by Mlambo et al[34] supports these findings by concluding that organisations should adequately make continuing professional development accessible for registered nurses to keep the staff up to date.

## Navigating obstacles and cultivating support

The results of the present study show experiences of staff shortage and time shortage, which hindered participants' ability to perform mentoring activities and participate fully in being a mentor. The participants expressed a need for education, training, meeting other mentors and the importance of relationships, ie, 'not doing it alone'. These results are confirmed by previous research that showed how education, training and time allocated for the mentor to interact with the mentee are prerequisites for organisations to benefit from mentoring.[31] Other studies showed that workload management and scheduling influence how mentoring can be conducted and showed that nurse managers expressed similar barriers to mentoring, such as competing priorities for time, and lack of training.[30 35]

Understaffing is related to emotional exhaustion and lower well-being in the workgroup. The higher workload from understaffing also leads to role ambiguity, ie, staff being unclear about their roles.[36] Results in our study confirm findings of role ambiguity, as participants described thoughts of ending the mentorship due to unclear definitions of the responsibility it entailed and their role as mentors. In addition, participants in the present study also described other challenges; needing structure, ie, clear leadership and direction, to optimise the mentoring. These findings can be interpreted as participants experiencing inequity, that their input in mentorship is unequal to the outcomes they receive, as described in *equity theory* by Adams.[37] Inequity can, according to Adams,[37] lead to frustration, job-dissatisfaction and a person leaving their position.

The results of our study both confirmed and expanded prior research. Even so, mentoring is still not a part of everyday routines in all healthcare units. The importance and benefits of mentoring are known, but it is still not facilitated sufficiently. According to Dall 'Alba,[18] a person's possibilities to act and be are constrained by specific situations, as well as their history, traditions and conditions. The current situation in healthcare is not ideal for fully developing mentoring. There is a vicious cycle, where the high turnover of registered nurses leads to difficulties in implementing mentoring that contributes to difficulties retaining registered nurses. A recent report from the International Centre on Nurse Migration showed that the demand for registered nurses is on the rise in several countries and estimated that there will be a need for 13 million registered nurses globally, in the next decade.[38] Other reports on the global shortage of

registered nurses[39 40] have concluded that organisations must create positive work environments that maximise registered nurses' health, safety and well-being, and improve or sustain their motivation to prevent further shortages. Registered nurses are attracted to work and remain working due to, for example, opportunities to develop professionally, gain autonomy and participate in decision-making.[39 40]

Moreover, the results of the present study show the frustration evident in the participating registered nurses' narratives. They did not feel valued by the organisation and experienced inequity. Therefore, they could not thoroughly add value to their mentorships or to the organisation in return. A systematic review of interventions that promote registered nurses' retention by Lartey et al[41] concluded that most interventions have focused on retaining newly qualified nurses. However, keeping experienced registered nurses in the profession is essential for providing high-quality care, ie, effective, safe, evidence-based and person-centred healthcare to meet the evolving needs of patients.[42] Organisations need to invest in experienced registered nurses to encourage them to remain in the profession, to provide high-quality care, and as mentors for new nurses.[6] Therefore, listening to the registered nurses' experiences and giving them opportunities to develop professionally and participate in decision-making about mentoring are essential factors in keeping them in the profession and for their well-being.

## Strengths and weaknesses of the study

Having a variety of backgrounds and understandings in the research team broadened the discussions and proved an advantage in our ability to understand the data. A strength is also that we described our pre-understanding, and our diversity of backgrounds that might prevent bias and achieve consensus through co-creation; as different researchers in a study can interpret the data differently.[25] The researchers speak Swedish and Norwegian, and our variety in native languages was a strength, for example, we could validate the transcripts and analysis by ensuring that nothing got lost in translation. A weakness of the study can be that the first author (SKH) analysed the interviews but did not conduct any of the interviews. It can be a disadvantage where part of the latent content, such as nonverbal behaviours and interactions, can be missed in the analysis. However, all other authors performed several of the interviews and were part of the analyses process.

## Clinical implications

This study brings a more profound understanding of registered nurses' perspectives of being a mentor. This study offers evidence as to how mentoring should be facilitated, thus contributing to highly competent registered nurses and safer patient care.

## Suggestions for further research

More research is required to fully explore and implement mentoring as a concept. In addition, more research

is needed to develop, test, and evaluate evidence-based mentorship programmes in healthcare units and to create sustainable mentoring solutions.

## CONCLUSIONS

Being a mentor is a complicated role for registered nurses. The mentoring role is beneficial, that is, positive and rewarding, if facilitated effectively. The role and responsibilities in the mentoring relationship need to be fully established and defined. A structured organisation is essential to promote mentoring, as is taking experienced registered nurses' views into consideration. The structure of mentoring in nursing should be designed and facilitated in co-creation with experienced registered nurses as mentors.

**Acknowledgements** Our deepest gratitude to all the participating registered nurses for generously sharing their narratives. We want to thank Johan Åhlin and Jonas Carlström for collecting data. We also want to thank Nord University and Umeå University for the support they provided, making it possible to carry out the study.

**Contributors** SKH: Data analysis, Writing – original draft, Writing – review and editing. AK: Data collection, Validation of analysis, Writing – review and editing. YH: Project administration, Supervision, Data collection, Writing – review and editing. FN: Supervision, Writing – review and editing. JRV: Data collection, Writing – review and editing. KB: Guarantor, Project administration, Supervision, Data collection, Validation of analysis, Writing – review and editing.

**Funding** This work was supported by Nordland County [20/1879-14].

**Competing interests** None declared.

**Patient and public involvement** Patients and/or the public were not involved in the design, or conduct, or reporting, or dissemination plans of this research.

**Patient consent for publication** Not applicable.

**Ethics approval** This study involves human participants and was approved by The Swedish Ethical Review Authority (Dnr 2020-06187) and the Norwegian Centre for Research Data (No. 148896). Participants gave informed consent to participate in the study before taking part.

**Provenance and peer review** Not commissioned; externally peer reviewed.

**Data availability statement** No data are available.

**ORCID iDs**
Stina Kallerhult Hermansson http://orcid.org/0000-0001-5639-8829
Fredrik Norström http://orcid.org/0000-0002-0457-2175

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
