## [Reviewer comments · BMJ Open]

ARTICLE DETAILS

TITLE (PROVISIONAL)	EXPLORING REGISTERED NURSES' PERSPECTIVES AS MENTORS FOR NEWLY QUALIFIED NURSES: A QUALITATIVE INTERVIEW STUDY
AUTHORS	Kallerhult Hermansson, Stina; Kasén, Anne; Hilli, Yvonne; Norström, Fredrik; Vaag, Jonas Rennemo; Bölenius, Karin

VERSION 1 – REVIEW

REVIEWER	Yip, Ka Huen Caritas Institute of Higher Education, School of Health Sciences
REVIEW RETURNED	21-Mar-2024

GENERAL COMMENTS	Thank you for the opportunity to review the manuscript. Overall, a current topic for a broader readership and further exploration of this topic to describe registered nurses' experiences of being a mentor for newly qualified nurses in northern Norway and northern Sweden. A few questions / comments and suggestions: In your review, you mentioned in page 3 Line 28-29 "Personal growth and professional development occur in stages that rely on constructive relationships and support from other professionals". To this reviewer, an important and relevant literature on this aspect may have been missed. One example includes "Yip, Y. C., Yip, K. H., & Tsui, W. K. (2021). The transformational experience of junior nurses resulting from providing care to COVID-19 patients: From facing hurdles to achieving psychological growth. International Journal of Environmental Research and Public Health, 18(14), 7383. doi: 10.3390/ijerph18147383". Consider acknowledging that paper or others to ensure an integral review of the concerned matter. In page 4 Line 19, cannot explain clear for any data saturation for participants sampling, relevant to the study. In page 4 Line 45-51, how to set the main interview questions, relevant to the study is not clear. In page 4-5 Line 59-15, unclear illustration how to have data analysis, relevant to the study is not clear. It's better to re-structure the methodological considerations in page 11 line 32-47 to page 4-5 line 59-15, for better explanation for data analysis. In page 6-7, for the subtheme: "feeling motivated in being a mentor", the authors could consider discussing whether there were
--

	any cultural differences between the Norwegian and Swedish participants regarding the personal characteristics and qualities they strived to embody as mentors. Since the findings included nurses from both Sweden and Norway, comparing their perspectives on desirable mentor attributes and behaviors could provide additional context about the range of qualities valued in an ideal mentor. In page 13 Line 23-29, the global shortage of registered nurses' issues, relevant to the study is not clear. In page 13 Line 38-39, what is high-quality care, relevant to the study is not clear.
--	--

VERSION 1 – AUTHOR RESPONSE

Reviewer comments:

1. In your review, you mentioned in page 3 Line 28-29 “Personal growth and professional development occur in stages that rely on constructive relationships and support from other professionals”. To this reviewer, an important and relevant literature on this aspect may have been missed. One example includes “Yip, Y. C., Yip, K. H., & Tsui, W. K. (2021). The transformational experience of junior nurses resulting from providing care to COVID-19 patients: From facing hurdles to achieving psychological growth. *International Journal of Environmental Research and Public Health*, 18(14), 7383. doi: 10.3390/ijerph18147383”. Consider acknowledging that paper or others to ensure an integral review of the concerned matter.

Thank you for your valuable suggestion. We have carefully read and considered referencing your article titled 'The transformational experience of junior nurses resulting from providing care to COVID-19 patients: From facing hurdles to achieving psychological growth.' We acknowledge the relevance of your work in exploring the experiences of junior nurses caring for COVID-19 patients. Even though our research objectives and context differ from those of your study, we believe that your study provides valuable perspectives for understanding our results. Therefore, we have added your article to our discussion on page 12.

2. In page 4 Line 19, cannot explain clear for any data saturation for participants sampling, relevant to the study.

Following your comment, we have added clarification on data saturation on page 4, under “Participants and sampling” (as suggested), and analysis saturation under “Data analysis”.

3. In page 4 Line 45-51, how to set the main interview questions, relevant to the study is not clear.

Thank you for this comment. In addition to information on the development of the interview guide, page 4, we have also added the interview guide as a supplementary file, following your comment.

4. In page 4-5 Line 59-15, unclear illustration how to have data analysis, relevant to the study is not clear.

We have revised the description of data analysis process for clarification, page 4.

5. It's better to re-structure the methodological considerations in page 11 line 32-47 to page 4-5 line 59-15, for better explanation for data analysis.

In accordance with your suggestion, the description of the methodological considerations have been moved to the methods section, page 5, and the heading "Reflexivity" have been changed to "Trustworthiness"

6. In page 6-7, for the subtheme: "feeling motivated in being a mentor", the authors could consider discussing whether there were any cultural differences between the Norwegian and Swedish participants regarding the personal characteristics and qualities they strived to embody as mentors. Since the findings included nurses from both Sweden and Norway, comparing their perspectives on desirable mentor attributes and behaviors could provide additional context about the range of qualities valued in an ideal mentor.

Thank you for this valuable comment. Although it would be interesting to investigate differences further, it is not within the scope of this study. The aim of this study was to describe the registered nurses experiences of being a mentor. In adapting the methodology of qualitative content analysis, it emphasizes variations in the text, the methodology does not entail comparisons between groups of participants (Graneheim, U. H., Lindgren, B. M., & Lundman, B. (2017). Methodological challenges in qualitative content analysis: A discussion paper. *Nurse education today*, 56, 29-34. <https://doi.org/10.1016/j.nedt.2017.06.002>).

The adapted method emphasizes the importance in finding participants that can provide variation, and rich descriptions of their lived experiences in the context of the study, therefore we sought after to include participants from both Sweden and Norway. We have revised the "Participants and sampling" section, for clarification, following your comment.

7. In page 13 Line 23-29, the global shortage of registered nurses' issues, relevant to the study is not clear.

Thank you for this helpful comment. For clarification, a sentence, and a reference, have been added on page 13.

8. In page 13 Line 38-39, what is high-quality care, relevant to the study is not clear.

For clarification, "high-quality care" is defined and explained in more detail, on page 13, following your comment.

Additional revisions by authors:

- Headings and subheadings have been edited in adherence to BMJ formatting guidelines, i.e., hierarchy of BOLD CAPS, bold lower case, Plain text, Italics.
- References edited into Vancouver style.
- For clarification, quotes have been edited into plain text (instead of italics) and quotation marks added.

VERSION 2 – REVIEW

REVIEWER	Yip, Ka Huen Caritas Institute of Higher Education, School of Health Sciences\
REVIEW RETURNED	02-May-2024
GENERAL COMMENTS	The revised manuscript sufficiently addresses the feedback and is now well-positioned for successful publication consideration given its rigorous methods, clear presentation for registered nurses.

\